# Influence of the Composition of the Hybrid Filler on the Atomic Oxygen Erosion Resistance of Polyimide Nanocomposites

**DOI:** 10.3390/ma13143204

**Published:** 2020-07-18

**Authors:** Olga Serenko, Ulyana Andropova, Nadezhda Tebeneva, Mihail Buzin, Egor Afanasyev, Aleksander Tarasenkov, Sergey Bukalov, Larisa Leites, Rinat Aysin, Lev Novikov, Vladimir Chernik, Ekaterina Voronina, Aziz Muzafarov

**Affiliations:** 1A.N. Nesmeyanov Institute of Organoelement Compounds of Russian Academy of Sciences, 28 Vavilova St., GSP-1, V-334, 119991 Moscow, Russia; hrh_uly@mail.ru (U.A.); buzin@ineos.ac.ru (M.B.); nambrot@yandex.ru (E.A.); buklei@ineos.ac.ru (S.B.); raman@ineos.ac.ru (L.L.); aysin@ineos.ac.ru (R.A.); aziz@ineos.ac.ru (A.M.); 2N.S. Enikolopov Institute of Synthetic Polymeric Materials of Russian Academy of Sciences, 70 Profsoyuznaya St., 117393 Moscow, Russia; tebeneva@mail.ru (N.T.); antarr@bk.ru (A.T.); 3D.V. Skobeltsyn Institute of Nuclear Physics, Lomonosov Moscow State University, 119991 Moscow, Russia; novikov@sinp.msu.ru (L.N.); vlachernik@yandex.ru (V.C.); voroninaen@nsrd.sinp.msu.ru (E.V.)

**Keywords:** polyimide composites, hybrid nanofillers, thermal properties, surface properties, atomic oxygen erosion

## Abstract

The structure and properties of nanocomposites based on organosoluble polyimide (PI) and branched functional metallosiloxane oligomers with different types of central metal atoms (Al, Cr, Fe, Zr, Hf and Nb) were investigated. Under the same weight content of the filler, the geometric parameters of the nanoparticles and thermal properties of the nanocomposites did not exhibit a direct relationship with the ability of the materials to withstand the incident flow of oxygen plasma. The atomic oxygenerosion resistance of the filled PI films was influenced by the composition of the hybrid fillerand the type of metal atom in the hybrid filler in the base metallosiloxane oligomer. To determine the effectiveness of the nanoparticles as protective elements of the polymer surface, the nanocomposite erosion yields pertaining to the concentration of the crosslinked organo–inorganic polymer forming the dispersed phase were determined and expressed in mmol per gram PI. The filler concentration in the polymer, expressed in these units, allows for comparison of the efficiency of different nanosize fillers for use in fabricating space survivable coatings. This can be important in the pursuit of new precursors, fillers for fabricating space survivable polymer composites.

## 1. Introduction

High-performance polymeric materials, for example polyimides, are extensively used in constructing spacecraft elements, in the form of coatings or binders of composite materials [1,2,3]. In space, the incoming atomic oxygen (AO) flow causes the erosion and oxidation of polymeric material surfaces, which considerably limits the spacecraft life [3,4,5,6,7,8,9,10,11,12].

To develop polyimides with a high space survivability by ensuring that the AO erosion yield is less than that of origin polyimide, silicon is introduced to improve the AO resistance (POSS blocks [13,14,15,16,17,18,19,20], hyperbranched polysiloxane [21,22], polysiloxane [23,24] and silica [25,26,27,28]). The erosion resistance of silicon containing polyimides to atomic oxygen is based on the formation of an inert silica protective layer, which prevents the polymer from eroding when exposed to AO and impedes the penetration of the atomic oxygen into the polymer inner layers [13,14,15,16,17,18,19,20,21,22,25,26,27,28]. A protective silica layer is formed after the ablation of the upper organic layer and oxidation of the Si–O–Si moiety to a SiO_2_ passivating layer [21,29,30,31].

Based on this mechanism, it is possible to identify techniques to increase the resistance of polymeric materials, especially that of polyimide (PI) coatings, to AO. It is necessary to increase the “quality” of the formed inert silica protective layer, which is determined by its surface density. This parameter depends on the silica–moietyconcentration and the associated distribution in the polymer. Thus, increasing the content of POSS blocks in PI can increase its resistance to AO [13,14,15,16,17,18,19,20].

The introduction of not only the Si–O–Si moiety (for example, in the form of POSS blocks) and SiO_2_ particles but also ZrO_2_ [32,33], TiO_2_ [34] or Al_2_O_3_ [35] particles into the PI can enhance the resistance of the PI materials to the action of AO. This mechanism of erosion protection against AO is also exploited when using PI filled with sol-gel through traditional precursors of the inorganic component of the composition (organofunctional silanes, metal alkoxides or their mixtures [36,37,38,39]). Although this approach can clarify the physical protective function of the filler, it does not take into account the role of the elemental composition (Si–O–Si, M–O–M, M=metal atom), and more specifically, the presence of the metal atoms in the inert protective layer. To experimentally examine this problem, branched functional metallosiloxane oligomers (BFMSOs) can be employed. BFMSOs have the general formula M[O-Si(R′)(OR″)_2_]_n_, where R′ and R″ are hydrocarbon substituents [40]. BFMSOs can be used with organosoluble PI to obtain coatings or films filled in situ under the same conditions by employing the “one step” technique because imidization of the polyamide acid/filler mixture is not required, and a high humidity is not required when introducing the catalyst into the reaction volume during the in situ polymer filling [41,42]. Furthermore, the particles formed in the polymer have a hybrid chemical structure and contain M–O–Si–O–Si bonds [42]. Therefore, using BFMSOs with different types of central metal atoms can allow the adjustment of the elemental composition of the inert protective layer.

In this study, the AO resistance of in situ filled polyimides developing using different BFMSOs as precursors of the dispersed phase was investigated by exposing the PI surface to various AO fluences, and the protection/oxidation mechanism was clarified. The presented findings can provide guidance for the development of a new effective strategy to fabricatepolymer coatings and films with excellent atomic oxygen resistant properties.

## 2. Materials and Methods

### 2.1. Materials

PI based on 4,4′-(9-fluorenylidene)dianiline and 3,3′,4,4′-diphenyl oxide tetracarboxylic acid has been used [43]. The molecule structure is given in Appendix A. The density of PI was 1.38 g/cm^3^ [42].

Tris-(methyldiethoxysiloxy)aluminum (below Al-MDES), tris-(methyldiethoxysiloxy)iron (Fe-MDES), tris-(3-aminopropyldiethoxysiloxy)chromium (Cr-MDES), tetrakis-(methyldiethoxysiloxy)zirconium (Zr-MDES), tetrakis-(methyldiethoxysiloxy)hafnium (Hf-MDES) and pentakis-(methyldiethoxysiloxy)niobium (Nb-MDES) have been used as precursors of the dispersed phase of the composites. Precursors were synthesized according to known methods [40]. The structural formulas of the compounds are given in Appendix A.

### 2.2. Preparation of Filled Films

Filled polymer films were prepared according to known method [42]. A detailed description of the preparation of filled films, andcompositions of films, are provided in Appendix A. The precursor concentration was 3 and 14 wt.%.

### 2.3. Material Characterization

FTIR spectra were recorded on a Tensor 37 (Bruker, Ettlingen, Germany) Fourier spectrophotometer with a resolution of 2 cm^−1^ for films placed on KBr plates. Attenuated total internal reflectance (ATR ) FTIR spectra were collected on a Vertex 70v (Bruker, Ettlingen, Germany) instrument using an ATR unit with a diamond element. The resolution was 2 cm^−1^.

Raman spectra in the region of 100–4000 cm^–1^ were registered on a laser Raman spectrometer, Jobin-Yvon LabRAM-300 (HORIBA, Tokyo, Japan) equipped with a sensitive charger – coupled device (CCD) detector and a microscope. The excitation line used was the 632.8 nm line of a He/Ne laser, its output not exceeding 3 mW; the frequency positional accuracy was ±2 cm^−1^. This installation allows for the micromapping of the sample surface with a resolution of 2 μm. The samples were scales or films of the compound placed on the microscope stage.

The microstructure of the samples was studied by transmission electron microscopy (TEM) using an LEO912 AB Omega microscope (Karl Zeiss, Oberkochen, Germany). Samples of 100–200 nm thickness for TEM were prepared using a RECHERT-JUNG ultramicrotome (Karl Zeiss, Oberkochen, Germany).

Surface morphologies of the samples were observed with a scanning electron microscope (SEM) JSM-7001F (JEOL, Tokyo, Japan).

Thermomechanical analysis was performed on a TMA Q400 (TA Instruments, New Castle, DE, USA) by scanning the temperature range from 15 to 700 °C at a heating rate of 5 °C/minand a static load of 1 N.

Thermal gravimetric analysis (TGA) was conducted on a Derivatograph-C instrument (MOM, Budapest, Hungary) in air (temperature range from room to 900 °C) at a heating rate of 10 °C/min; the sample weight was ~10 mg.

### 2.4. AO Beam Exposures

Graund-based atomic oxygen exposures of polymer films were performed in a magnetoplasmadynamic accelerator developed at the V.D. Skobeltsyn Institute of Nuclear Physics, Lomonosov Moscow State University, Moscow, Russia [44]. Film samples 20×20 mm^2^ in size were used. The specimens were degassed beforehand and held for 24 h at a temperature of 20 °C in a vacuum of 10^−4^ Pa. A detailed description of the method is provided in Appendix A.

## 3. Results

### 3.1. Chemical Structure of the Fillers and Composites

The formation of the filled polymer through hydrolysis and condensation reactions, in which the BFMSO precursor is transformed into a highly crosslinked solid, is illustrated in Appendix A.

Previously, it was shown that no chemical interactions occur between the PI and fillers based on Al-, Zr- or Nb-MDES and no hydrogen bonds are formed [42]. In the present work, the similar research technique was used for testing the films involving PI-Fe-MDES and PI-Cr-MDES.

We studied the changes in the vibrational spectra after filling PI with the mentioned metalalkoxysiloxane precursors with a concentration of 14 wt.%. The ν_CH_ region of the Raman spectra for the PI, PI-Fe-MDES films and Fe-MDES is shown in Figure 1a. The presence of the filler in the composites wasconfirmed by the appearance of the new bands in the Raman spectra, at ~2910 and ~2970 cm^−1^, corresponding to the methyl groups in the Si atoms (ν_CH3_). The γ-aminopropyl group in the composite PI-Cr-MDES was preserved, as indicated by the Raman spectrum shown in Figure 1b (ν_CH_ bands at ~2890 and ~2925 cm^−1^ and ν_NH_ broad band near ~3215 cm^−1^). The other vibrational bands of the filler with the polar skeletal bonds in the Raman spectra of the composites were weaker in comparison to the intense PI Raman spectrum and were therefore not detectable.

In the IR spectra shown in Figure 2, it can be observed that the region of the ν^as^_MOSiOSi_ bands of the fillers overlap with the PI bands. The difference in the IR spectrum of the broad band of the complex contour in the 900–1100 cm^−1^ region, corresponding to the ν^as^_MOSiOSi_ modes, could be determined only using computerized spectra subtraction. Figure 2 shows the results of this subtraction procedure for the IR spectra of the composites of PI with Fe-MDES. The analogous IR results were obtained for the composites of PI with Cr-MDES (see Appendix A). The correctness of the subtraction was verified by observing the straight line in the difference spectra, observed in the place of the PI bands (1250–1800 cm^−1^).

The results of the vibrational spectra investigations of the composites thus proved that the product of the Fe-MDES and Cr-MDES precursor hydrolytic polycondensation in the PI volume contains the methyl and γ-aminopropyl group, respectively. M–O–Si and Si–O–Si units present in the fillers were based on the different metalalkoxysiloxane precursors. The PI frequencies in the spectra of the composites were similar to those in the spectrum of the initial PI. Consequently, the fillers based on Fe-MDES and Cr-MDES were inert to the PI.

### 3.2. Morphology of Filled PI Films

The TEM analysis of the samples indicated that, in all the cases, the in situ filling of the PI produced nanocomposites.The sizes of the generated dispersed phase particles were in the nanometer range, depending on the type of BFMSO used.

Figure 3 shows the structures of the filled films developed using a 14 wt.% precursor. The particle sizes in the films involving PI-Al-MDES and PI-Fe-MDES were 11 ± 2 and 10 ± 4 nm, respectively (Figure 3a,b). In the PI-Cr-MDES sample, the particles were smaller, and their size did not exceed 2 ± 1 nm (Figure 3c). The use of Zr-MDES (Figure 3d) led to the formation of nanoparticles with a diameter of 8 ± 3 nm, whereas diameters of 5 ± 2 nm and 5 ± 1 nm were attained when Hf-MDES and Nb-MDES were used, respectively (Figure 3e,f). In the films composed of PI-Zr-MDES and PI-Hf-MDES, the particles could combine to form irregular elongated nanoscale aggregates.

Thus, when using Cr-MDES as a precursor of the dispersed phase, the particles in the PI film were the smallest; the size of particles obtained using Al-MDES or Fe-MDES was five times larger.

### 3.3. Thermal Properties

Table 1 lists the temperatures characterizing the thermo-oxidative stability of the PIs, fillers, and nanocomposites based on the temperature of the onset of the decomposition, T_d_; the temperature at which the sample exhibits a 5% weight loss, T_5%_; and the temperature at which the thermo-oxidative degradation of the matrix occurs, T_dm_. The values of T_d_ and T_5%_ of the fillers were lower than the corresponding PI values. Apparently, filling the polymer with particles with lower T_d_ and T_5%_ than those of the matrix polymer led to a decrease in the T_5%_ values of the filled systems.

The precursor type determines the thermal oxidation resistance of the matrix polymer in the composites. For samples composed of PI-Al-MDES, PI-Zr-MDES, PI-Hf-MDES and PI-Nb-MDES, the T_dm_ values were the same and coincided with the temperature of the thermo-oxidative destruction of the initial PI. The T_dm_ values of PI-Fe-MDES and PI-Cr-MDES films were less than those of the unfilled PI. The fillers based on Fe-MDES exhibited the least resistance to the thermal-oxidative degradation of the matrix polymer. In this case, the values of T_dm_ were lower than those of the initial polymer by 77–85 °C. The decomposition products of the fillers in PI-Fe-MDES and PI-Cr-MDES samples likely initiated and/or contributed to the matrix thermal degradation.

Table 1 lists the obtained values for the glass transition temperatures T_g_ of the PI-based samples. The T_g_ values of the filled PI exceeded the temperature of the onset of the thermo-oxidative degradation of the filler (T_d_); however, these values were lower than the corresponding matrix polymer temperature (T_dm_). Therefore, the matrix devitrification was either accompanied by the destruction of the filler or it occurred at the end of the thermo-oxidative decomposition. By comparing the T_g_ values of the filled systems with those of the initial PI, we can conclude that the filler and its decomposition products do not adversely influence this characteristic of the composite. Therefore, when using Al-MDES as a precursor, the glass transition temperatures of the composites do not depend on the content of the dispersed phase and exceed the T_g_ of the initial PI by 9 °C.

The glass transition temperature for the PI-Zr-, Hf-, Nb- and Fe-MDES systems increased. In the Fe-MDES, the temperature increased by 12–14 °C. The destructive processes of these fillers in the polymer at elevated temperatures likely promoted the immobilization of the portion of the matrix polymer in contact with the particles of the dispersed phase, which led to an increase in the glass transition temperature of these systems.

The glass transition temperature of the PI-Cr-MDES composites changed only slightly with the introduction of 3 wt.% Cr-MDES and decreased by 10 °C when its concentration was increased to 14 wt.%. Among the precursors, the chemical structure of Cr-MDES includes the (-(CH_2_)_3_-NH_2_) group. According to Chen X.Y et al. [45], the amine group of the precursor (filler) at high temperatures chemically reacted with the imide group of the matrix polymer. The proposed reaction scheme is shown in Appendix A. The occurrence of this reaction, namely, the opening of the imide cycle in a part of the monomer units of the polymer, led to a decrease in the chain stiffness, thereby resulting in a decrease in the glass transition temperature of PI [45]. This reaction likely led to the observed decrease in the glass transition temperature of the filled PI-14 wt.% Cr-MDES film.

### 3.4. AO Erosion

According to the results described in [42], the effect of the type of BFMSO used on AO erosion resistance of PI based nanocomposites is notable when the content of this precursor is more than 3 wt.%. In the present work, films obtained using 14 wt.% BFMSO were irradiated with AO.

Figure 4 shows the dependence of the specific mass loss (Δm/S) on the AO fluence (F) of Kapton H films (reference sample), the initial organosoluble PI, and filled films obtained using different precursors. The dependence for the organosoluble PI and the reference sample is linear; however, the PI exhibited a greater resistance to AO than that of the reference sample. In situ filling of the polymer led to a sharp decrease in the erosion of the samples under the influence of AO. An analysis of the experimental dependence indicated that the samples could be divided into three groups, in which the Δm/S values of the samples differ by no more than 20%: (PI-Al-MDES and PI-Cr-MDES), (PI-Fe-MDES and PI-Zr-MDES), (PI-Hf-MDES and PI-Nb-MDES). The Δm/S values at F= 7.7×10^20^ O atom/cm^2^ were 5.5 and 7.6, 11.5 times less than those for the unfilled PI for the first, second and third groups of the filled films, respectively.

Figure 5a shows the dependence of the erosion yields (E_y_) of Kapton H and organosoluble PI, and Figure 5b shows the relative erosion yields of the composites (E_rel_=E_yc_/E_yp_, where E_yc_ and E_yp_ denote the erosion yields of filled and unfilled PI, respectively) on the fluence AO. For the unfilled polymers, the E_y_ values did not change with increasing AO fluence; the values were 3.0 × 10^−24^ cm^3^/atom and 2.4 × 10^−24^ cm^3^/atom for the Kapton and PI, respectively. The dependence of E_rel_ on the AO fluence determined using the experimental data of Δm/S on F could also be divided into three groups. For the first (PI-Al-MDES and PI-Cr-MDES), second (PI-Fe-MDES and PI-Zr-MDES) and third (PI-Hf-MDES and PI-Nb-MDES) groups of the filled PI films, the E_rel_ values decreased to 18%, 13% and 8% with increasing AO fluence. Therefore, filling the polymer with particles composed of BFMSO can reduce the erosion yield by 82–92%.

Figure 6 shows the SEM images of the samples after exposure to AO. The surface of the irradiated PI film exhibited the “carpet” structure characteristic of the polymer. The filled films after exposure to AO exhibited a different morphology. Specifically, microcracks were observed on the surface of samples based on PI and Al-, Cr-, Fe-, Zr-MDES, the opening of which did not exceed 0.5 μm. The PI-Hf-MDES and PI-Nb-MDES films were less damaged. In addition, areas with and without microcracks were observed on the surface. The maximum size of these regions in PI-Nb-MDES was approximately 3 × 5 μm^2^. An inhomogeneous surface structure was formed under the influence of AO due to the different rates of the destruction processes; specifically, the erosion rate of polyimides filled with particles composed of Hf- or Nb-MDES in different local zones of the surface was significantly different. This aspect is the likely reason for the higher resistance of these composites to AO compared to the other films.

## 4. Discussion

The “protective function” of the filler in the polymer composition is determined by the size of its particles. As discussed, the particle size in the PI increased in the following range in terms of the precursors: Cr-MDES < (Hf-MDES, Nb-MDES) < Zr-MDES < (Al-MDES, Fe-MDES). A discrepancy can be noted when comparing this series with the decreasing order of the relative erosion coefficients of the samples: (PI-Hf-MDES и PI-Nb-MDES) < (PI-Fe-MDES и PI-Zr-MDES) < (PI-Al- MDES, PI-Cr- MDES). In the last group (PI-Al-MDES, PI-Cr-MDES), the particle size was different by a factor of 5; however, the values of E_rel_ were equivalent. Moreover, although the particles of the films involving PI-Fe-MDES and PI-Zr-MDES had different sizes, the E_rel_ values were similar. Therefore, the size of the dispersed phase nanoparticles did not considerably influence the erosion resistance of the filled films. In addition, although the diameters of the nanoparticles in PIs depended on the type of precursor used, the difference was only minimal and not by an order of magnitude.

The factors that determine the resistance of a material to AO do not necessarily influence the resistance to the thermal oxidative degradation. As discussed, the decomposition products of the fillers based on Cr-MDES and Fe-MDES negatively influenced the heat resistance of the matrix polymer. However, the erosion resistance of the PI-Cr-MDES film was similar to that of the PI-Al-MDES sample, and thus the presence of filler particles in the PI did not affect the onset temperature of decomposition of the matrix polymer. A similar phenomenon was observed for PI-Fe-MDES and PI-Zr-MDES films.

Thus, the geometric parameters of the filler and thermal properties of nanocomposites do not exhibit a direct relationship with the ability of the material to withstand the incoming flow of oxygen plasma.

The effectiveness of the filler in protecting the polymer depends on the number of “protective elements” in the PI and their composition. The “number of protective elements” of a PI film can be considered as the number of moles of a net organo–inorganic polymer that forms the dispersed phase in PI. Thus, under the same weight content of the filler, the “amount of substance” of the dispersed phase based on the studied precursors is different due to the different molar masses of the monomer units of the crosslinked organo–inorganic polymer (see Appendix A). Appendix A lists the filler concentration in the PI, expressed in units of mmol of filler per gram of PI (mmol/g). Figure 7 presents the erosion yields of the nanocomposites as a function of the filler concentration, expressed in mmol/g. At similar particle weights, the concentrations of the filler in mmol/g vary. Thus, for films composed of PI-Al-MDES and PI-Nb-MDES, the filler concentrations in mmol/g differed by 1.5 times. As shown in Figure 7, each of the studied PI-BFMSO systems contained a different number of moles per gram of PI, and consequently, the number of “protective elements” and the effect of the composition of the hybrid filler and the nature of the central metal atom of the precursor on the erosion resistance of the nanocomposites were different. For example, a high protection efficiency, characterized by a low erosion yield of the nanocomposite, could be achieved with fewer “protection elements” of the filled film when using the dispersed phase Nb-MDES as a precursor; the corresponding efficiency when using Al-MDES to decrease the erosion resistance of PI coatings was lower. With an increase in the number of “protective elements” of the polymer by 1.5 times, the erosion yield of the nanocomposite increased by 2.5 times. Consequently, the filler concentration in the polymer, expressed in units of mmol of filler per gram of the polymer, allowed for comparison of the efficiency of different nanosize fillers using for fabricating space survivable coatings. This is particularly important in the pursuit of new precursors, fillers and polymer composite materials for space applications, or in the proposed use of untested materials.

Summarizing the obtained experimental results, the following conclusions can be drawn. The protective function of nanoparticles is determined by its hybrid structure as a whole, and not by individual blocks (M–O–Si or Si–O–Si blocks). Earlier we assumed the following mechanism for the protective action of the filler formed in situ: preventing the penetration of atomic oxygen into the inside layers of the polymer; reinforcing the surface layer and preventing material erosion during AO irradiation; chain breaking of the radical process of polymer oxidative destruction due to the death of the formed radicals on the nanoparticle surfaces; and self-healing of the protective layer under the action of AO during the post-reaction hydroxy-group condensation, the catalyst of which is the metal atom in the crosslinked organo–inorganic polymer composition [42]. However, additional studies are required to verify these assumptions.

## 5. Conclusions

The use of BFMSOs as precursors of the PI-dispersed phase does not impair its unique thermal properties and increases the glass transition temperature. Filling the polymer with hybrid particles improves its space survivability. The relative erosion yields of the composites at a fluence of 7.60 × 10^20^ atom O/cm^2^ are less than that of the initial polymer by more than 80–90%.This aspect indicates that the in situ filled polyimides obtained using BFMSOs as the dispersed phase precursors can be used as surface protective materials onboard spacecraft to resist AO attack in the low earth orbit environment.

Under the same organosilicon frame of the central metal atom in the hybrid filler, the type of metal atom affects the erosion resistance of the filled PI films exposed to AO. Considering that different central atoms of the filler (precursor) lead to different atomic oxygen erosion resistances of the nanocomposites for the same weight content of the filler, metallosiloxane oligomer precursors exhibit promising potential to be used to develop functional polymer materials that are not only resistant to AO but also exhibit other useful properties (mechanical strength, impact resistance, dielectric properties, etc.).

## Figures and Tables

**Figure 1 materials-13-03204-f001:**
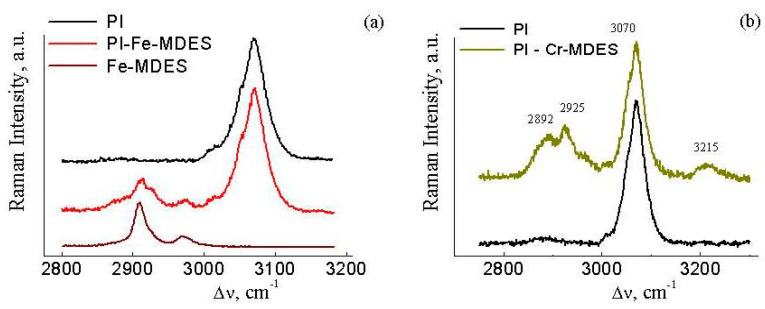
The region of ν_CH_ modes in the Raman spectra of polyimide (PI), the composite PI-Fe-MDES, the filler based on Fe-MDES (**a**) and PI-Cr-MDES, the filler based on Cdr-MDES (**b**).

**Figure 2 materials-13-03204-f002:**
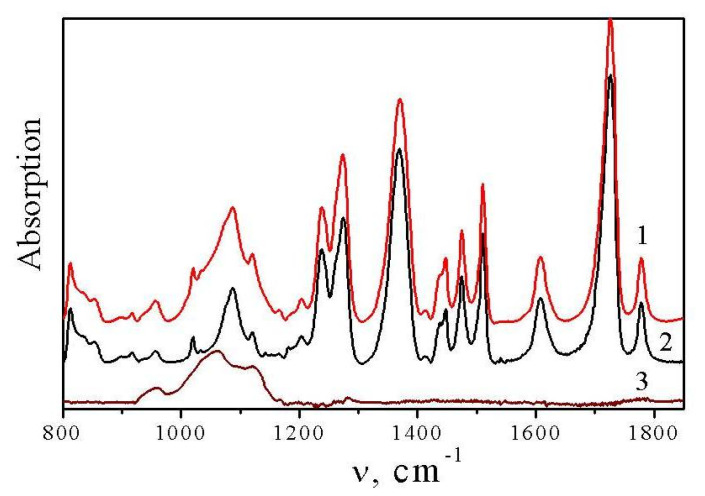
IR spectra of the films (on KBr plates) in the region 800–1800 cm^−1^ of PI-14 wt.% Fe-MDES (1), PI (2) and the difference spectrum (3), proving the presence of Fe-MDES in the composite (a broad band 900–1150 cm^−1^ in the difference spectrum).

**Figure 3 materials-13-03204-f003:**
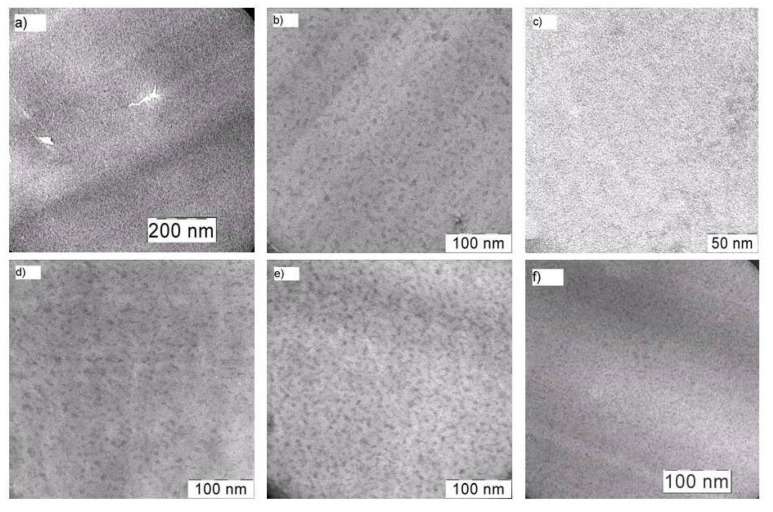
TEM images of nanocomposites on base PI and metalloalkoxysiloxane precursors: Al-MDES (**a**), Fe-MDES (**b**), Cr-MDES (**c**), Zr-MDES (**d**), Hf-MDES (**e**) and Nb-MDES (**f**). Precursor concentration is 14 wt.%.

**Figure 4 materials-13-03204-f004:**
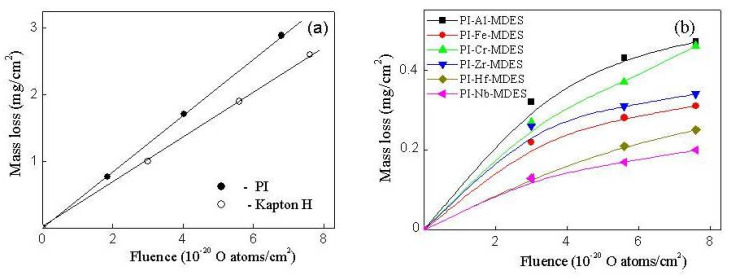
Dependences of the mass loss on the atomic oxygen (AO)fluence of Kapton H film (reference sample), the initial organosoluble PI (**a**) and PI nanocomposites obtained using different precursors (**b**). Precursor concentration is 14 wt.%.

**Figure 5 materials-13-03204-f005:**
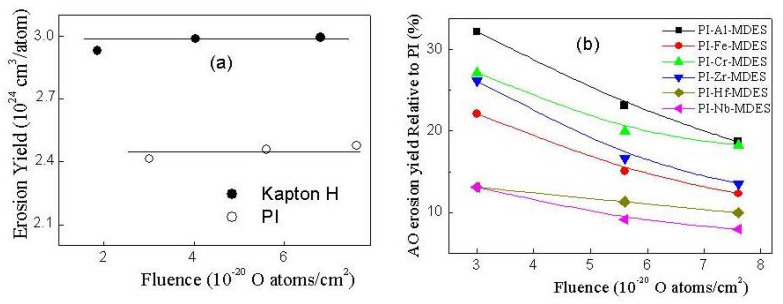
Dependences of the erosion yields of Kapton H (reference sample), the initial organosoluble PI (**a**) and the relative erosion yields of PI nanocomposites obtained using various precursors (**b**) on the fluence AO. Precursor concentration is 14 wt.%.

**Figure 6 materials-13-03204-f006:**
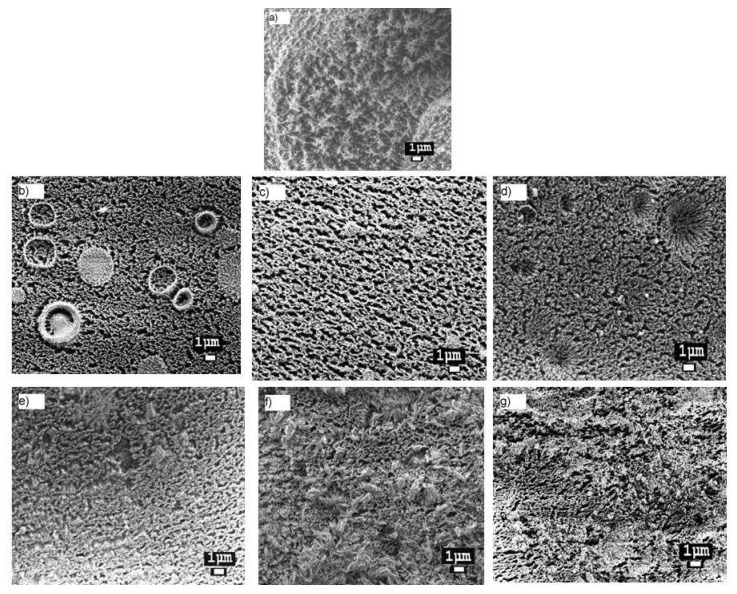
SEM images of samples after exposure to AO: (**a**) PI, (**b**) PI-Al-MDES, (**c**) PI-Cr-MDES, (**d**) PI-Fe-MDES, (**e**) PI-Zr-MDES, (**f**) PI-Hf-MDES and (**g**) PI-Nb-MDES. Precursor concentration is 14 wt.%.

**Figure 7 materials-13-03204-f007:**
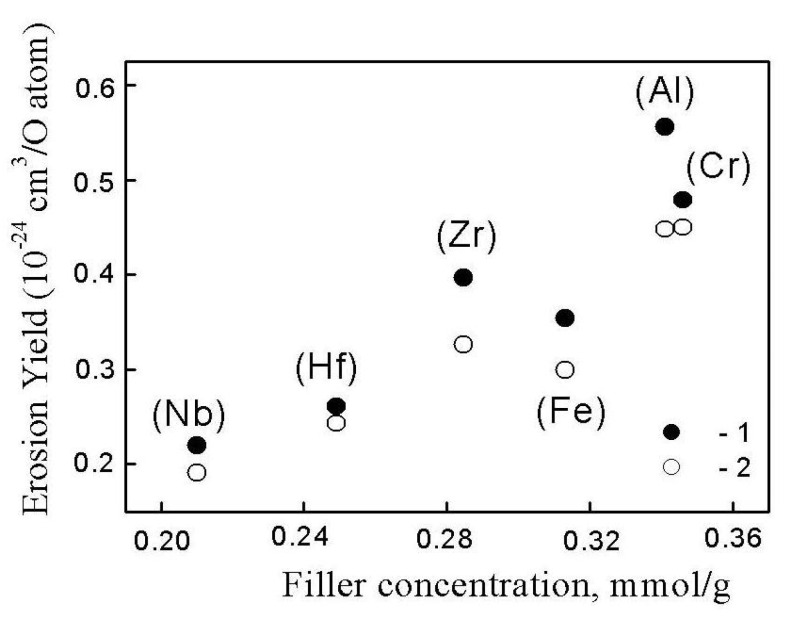
Erosion yields of PI nanocomposites obtained using ***M***-MDES precursors (***M*** = Al, Fe, Cr, Zr, Hf and Nb) on the filler concentration. The AO fluence is 5.6 × 10^20^ (1) and 7.6 × 10^20^ atoms O/cm^2^ (2).

**Table 1 materials-13-03204-t001:** Thermal properties of PI, metalloalkoxysiloxanes and nanocomposites based on them *.

Sample	The Precursor Concentration (wt.%)	T_g_ (°C)	T_d_ (°C)	T_5%_ (°C)	T_dm_ (°C)	Solid Residue (wt.%)
PI	0	381	497	570	-	0
Al-MDES	100	-	268	310	-	86
Fe-MDES	100	-	235	290	-	80
Cr-MDES	100	-	271	367	-	56
Zr-MDES	100	-	278	362	-	87
Hf-MDES	100		247	416		90
Nb-MDES	100	-	259	334	-	86
PI-Al-MDES	3	390		538	497	13
	14	390		456	497	6
PI-Fe-MDES	3	395		476	415	8
	14	389		450	420	14
PI-Cr-MDES	3	386		480	457	4
	14	371		407	445	7
PI-Zr-MDES	3	387		550	497	10
	14	386		507	497	17
PI-Hf-MDES	3	387		548	497	8
	14	393		573	497	18
PI-Nb-MDES	3	381		566	497	12
	14	386		572	502	13

* T_g_—the glass transition temperature; T_d_—the temperature of the onset of the decomposition; T_5%_—the temperature at which the sample exhibits a 5% weight loss; T_dm_—the temperature of the onset of thermo-oxidative degradation of the matrix polymer.

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
