# Peer review of "Influence of the Composition of the Hybrid Filler on the Atomic Oxygen Erosion Resistance of Polyimide Nanocomposites"

_materials, 2020, doi:10.3390/ma13143204_

Round 1
Reviewer 1 Report
The manuscript is of high practicale value.
Please check Fig. 3, which is incomplete in the copy available (pdf).
Author Response
Reviewer 1
Please check Fig. 3, which is incomplete in the copy available (pdf).
Fig. 3 is corrected.
Reviewer 2 Report
The authors reported a stuyd of the atomic oxygen erosion resistance of polyimide with addition of metallosiloxane. The use of the silicon, in the form of POSS and polysiloxane has been extensively in the past.The addition of some metal oxides can enhance the resistance of the PI. However, in the paper, the importance of using metallosiloxane is poored addressed. There are several key concerns that should be answered: 1. How is the resistance performace of metallosiloxane functionlized PI compared with previsous examples (fore example POSS-PI, SiO2-PI and ZrO2-PI)? 2. What is the influence of metallosiloxane to the PI film, in terms of the mechanical strength and optical transparency? 3 How to quantitatively characterize the metal element? The elemental composition of PI composite can be obtained by XPS analysis. 4. If the metallosiloxane enhance the resistance performace, what is the protecting mechanism? Lastly, the figure 3b-e are missing. In summary, I can't recommend it to be published in its current status, but I am happy to review it after those problems are addressed.
Author Response
Answers are given in the attached file.

Reviewer 3 Report
The presented manuscript contains interesting results of an actual study related to the development of composite materials that can be used in outer space (space program).
The article, of course, thematically suits the topic of the Materials.
However, it may be published in the journal after some changes.
On the first page, you need to remove typos in the line of authors.
Fig. 3 needs to be replaced because it is not readable.
Lines 180 - 181: This conclusion requires clarification. Is this an assumption?
Fig. 4a: The legend is surprising. Isn't Kapton a polyimide?
Fig. 7 is little informative and not required in the article. It does not answer a simple question - how do different precursors affect Erosion yields at same concentrations. Without this discussion, all discussions of the data in Fig. 7 seem unreasonable.
The Supplementary Materials: need to improve the English language.
The polyimide formula with the name assigned to it, as well as the precursor formulas, should be transferred to the main file of the article.

Author Response
Reviewer 3
- On the first page, you need to remove typos in the line of authors.
It is done.
- Fig. 3 needs to be replaced because it is not readable.
Fig. 3 is corrected.
3.Lines 180 - 181: This conclusion requires clarification. Is this an assumption?
Сorrection was made: “Apparently, filling the polymer with particles with lower Td and T5% than those of the matrix polymer leads to a decrease in the T5% values of the filled systems”.
- Fig. 4a: The legend is surprising. Isn't Kapton a polyimide?
A title of Fig. 4 is corrected: “Dependences of the mass loss on the AO fluence of Kapton H film (reference sample), the initial organosoluble PI (a) and PI nanocomposites obtained using different precursors (b). Precursor concentration is 14 % wt.”
- 5. Fig. 7 is little informative and not required in the article. It does not answer a simple question - how do different precursors affect Erosion yields at same concentrations. Without this discussion, all discussions of the data in Fig. 7 seem unreasonable.
Simple questions are the most difficult. The results presented in fig. 7 allow to conclude, that protective function of nanoparticles is determined by its hybrid structure as a whole, and not by individual blocks (M-O-Si- or Si-O-Si- blocks).
So, we do not agree with your recommendation.
Earlier we put forward several assumptions: “The protective action mechanism of the in situ formed filler was proposed: preventing the penetration of atomic oxygen into the inside layers of the polymer; reinforcing the surface layer and preventing material erosion during AO irradiation; chain breaking of the radical process of polymer oxidative destruction due to the death of the formed radicals on the nanoparticle surfaces; and self-healing of the protective layer under the action of AO during the post-reaction hydroxy-group condensation.” (Andropova, U.; Serenko, O.; Tebeneva, N.; Tarasenkov, A.; Buzin, M.; Afanasyev, E.; Sapozhnikov, D.; Bukalov, S.; Leites, L.; Aysin, R.; Polezhaev, A.; Naumkin, A.; Novikov, L.; Chernik, V.; Voronina, E.; Muzafarov, A. Atomic oxygen erosion resistance of polyimides filled hybrid nanoparticles. Polymer Testing 2020, 84, 106404). Each of them is legitimate, but requires experimental evidence.
- The Supplementary Materials: need to improve the English language.
It is done.
- The polyimide formula with the name assigned to it, as well as the precursor formulas, should be transferred to the main file of the article.
The polyimide, as well as the precursor formulas were given in our articles earlier (Andropova, U.S.; Tebeneva, N.A.; Serenko, O.A.; Tarasenkov, A.N.; Buzin, M.I.; Shaposhnikova, V.V.; Muzafarov, A.M. Nanocomposites based on polyarylene ether ketones from sol–gel process: Characterizations and prospect applications. Mater. Des. 2018, 160, 1052–1058; Andropova, U.; Serenko, O.; Tebeneva, N.; Tarasenkov, A.; Buzin, M.; Afanasyev, E.; Sapozhnikov, D.; Bukalov, S.; Leites, L.; Aysin, R.; Polezhaev, A.; Naumkin, A.; Novikov, L.; Chernik, V.; Voronina, E.; Muzafarov, A. Atomic oxygen erosion resistance of polyimides filled hybrid nanoparticles. Polymer Testing 2020, 84, 106404). According to existing rules, we can’t give them in the main text of the paper now. Therefore they are given in Supplementary Materials.
Reviewer 4 Report
Please attach Figure 3 (b, c, d, e). These figures are completely missing and it is impossible to follow the explanation of the TEM analysis.
In general, the presentation of the manuscript is very good and close to the point of the main idea of the work done. The authors have used several complementary techniques are used and the results are interesting.
Author Response
Reviewer 4
Please attach Figure 3 (b, c, d, e). These figures are completely missing and it is impossible to follow the explanation of the TEM analysis.
Fig. 3 is corrected.
Reviewer 5 Report
This article deals with atomic oxygen erosion resistance of polyimide nanocomposites. In particular, nanocomposites based on organosoluble polyimide and branched metallosiloxane oligomers with different types of central metal atoms were investigated. The characterization was performed by means of microscopy and Raman and FTR spectroscopy. The authors conclude that by filling the polymer with hybrid particles improves its space survivability. The manuscript appears interesting and well written and may suitable for publication. See below for some suggestions.
-The methods section could be improved with further details.
-The precise composition of each sample tested should be clearly reported from beginning (a new table, acronyms?).
-Figure 3 should be replaced, and in any case the authors should well explain the significance for each sample of this TEM investigation. If there are no relevant differences, may this figure should be eliminated.
-Fig 4 should be better explained, you have panel a and b…..
Author Response
Reviewer 5
1.The methods section could be improved with further details.
Detailed descriptions of the preparation of filled films, compositions of films are added in Supplementary Materials.
The precise composition of each sample tested should be clearly reported from beginning (a new table, acronyms?).
Compositions of films are provided in Table S1. Used abbreviation are reported in text of the paper.
Figure 3 should be replaced, and in any case the authors should well explain the significance for each sample of this TEM investigation. If there are no relevant differences, may this figure should be eliminated.
Fig. 3 is been corrected.
Fig 4 should be better explained, you have panel a and b…..
A title of Fig. 4 is corrected: “Dependences of the mass loss on the AO fluence of Kapton H film (reference sample), the initial organosoluble PI (a) and PI nanocomposites obtained using different precursors (b). Precursor concentration is 14 % wt.”
Round 2
Reviewer 2 Report
In author's reply, they stated that the protection mechanism of Metalloalkoxysiloxanes is still unknown and they believe it is too earlier to give possible explanation. This important point shall still be discussed and make a clear statement in the revised paper. In addition, the authors provide the date of elemental analysis from previous papers. However, the key to this is the metal concentration inside the material that is not characterized or discussed in the paper. Thus, I would suggest the authors to further revise their manuscript before publishing.
Author Response
In author's reply, they stated that the protection mechanism of Metalloalkoxysiloxanes is still unknown and they believe it is too earlier to give possible explanation. This important point shall still be discussed and make a clear statement in the revised paper. In addition, the authors provide the date of elemental analysis from previous papers. However, the key to this is the metal concentration inside the material that is not characterized or discussed in the paper. Thus, I would suggest the authors to further revise their manuscript before publishing.
The answer is given in the attached file.

Round 3
Reviewer 2 Report
The authors made corrections per reviewers suggestion. I would recommend to publish.